# Photothermal Effect and Phase Transition in VO_2_ Enhanced by Plasmonic Particles

**DOI:** 10.3390/ma16072579

**Published:** 2023-03-24

**Authors:** Vladimir Kaydashev, Boris Khlebtsov, Maxim Kutepov, Anatoliy Nikolskiy, Alexey Kozakov, Alexey Konstantinov, Alexey Mikheykin, Gevork Karapetyan, Evgeni Kaidashev

**Affiliations:** 1Laboratory of Nanomaterials, Southern Federal University, 200/1 Stachki Ave., 344090 Rostov-on-Don, Russia; 2Institute of Biochemistry and Physiology of Plants and Microorganisms RAS, Saratov Scientific Center, 13 Entuziastov Ave., 410049 Saratov, Russia; 3Institute of Physics, Southern Federal University, 194 Stachki Ave., 344090 Rostov-on-Don, Russia; 4Physics Faculty, Southern Federal University, 5 Zorge St., 344090 Rostov-on-Don, Russia

**Keywords:** vanadium dioxide, VO_2_, photthermal effect, surface acoustic waves, SAW, nanostars, coupled plasmonic nanoparticles

## Abstract

Phase change metasurfaces based on VO_2_, which are pre-heated with electric current and optically addressed by projected structured light hologram, are considered to become a new paradigm in programmed THz/middle IR flat optics. Macroscopic quasi-homogeneous arrays of Au nanoparticles show large near IR absorption and a significant photothermal effect capable of boosting a light-triggered switching of VO_2_ and are to be carefully examined. We propose a new approach to simultaneously probe the altered temperature and electric conductivity of a hybrid Au particle-VO_2_ film composite by monitoring a phase shift and attenuating a surface acoustic wave in a YX128° cut LiNbO_3_ substrate. The method shows a temperature resolution of 0.1 °C comparable with the best existing techniques for studying nanoobjects and surfaces. The laser-induced photothermal effects were characterized in a macroscopic array of Au nanostars (AuNSts) with different surface coverage. In a monolayer of 10 nm Au, coupled plasmonic nanoparticles were deposited on the LiNbO_3_ substrate. An optically triggered insulator-metal transition assisted by photothermal effect in AuNSts/VO_2_/TiO_2_/LiNbO_3_ composites was studied at varied light power. We believe that the proposed SAW-based method is of significant importance for the characterization and optimization of radiation absorbing or/and electrically heated elements of metasurfaces and other devices for lab-on-chip and optical communication/processor technology.

## 1. Introduction

Vanadium dioxide is considered one of the most promising materials with isolator-metal transition (MIT) to alter the phase and amplitude of passing or reflected middle IR, THz/GHz, and near IR waves [1,2]. VO_2_ metasurfaces are considered effective tools to in-situ manipulate a radiation wavefront by thermal, electrical or optical signals to focus radiation [3], obtain beam steering [4], alter polarization [3,5], filter wavelength [6] or achieve ultrafast modulation [7] in a tunable manner. Also, VO_2_ metasurfaces are prospective for developing lab-on-chip technology, optical chip-based self-trained neural networks and optical processing technologies. Recent advances in metasurfaces for manipulation of THz wavefront as well as the progress in VO_2_-based devices, are summarized in reviews [2,8,9].

We recently demonstrated a hybrid optoelectric approach for controlling VO_2_-based THz metasurface by pre-heating its pixels with electric current and switching them with laser light intensity as low as 0.3 W/cm^2^ in near IR or UV range [10]. The approach allows one to address individual pixels with light holograms generated by a digital micromirror device (DMD). We believe that metasurfaces with pixels that are optically addressed by projected structured light holograms will soon become a new paradigm in mid-IR/THz flat optics and photonics.

Plasmonic particles and nanosystems are well established as efficient light-to-heat nano-converters [11,12,13,14]. They are widely used in advanced healthcare for the selective killing of harmful cells in tissues and blood [15], as well as in highly efficient light absorbers [16] and thermophotovoltaics [17].

Recently we showed that plasmonic particles deposited on the surface of VO_2_ might boost the performance of light-triggered MIT [10]. Namely, a laser-induced switching threshold was reduced by ~30% when the VO_2_ metasurface was covered with a monolayer of Au particles. However, the switching efficiency of VO_2_, which is activated by a plasmonic particle, depends not only on the absorption of the plasmonic system but also on many features of a VO_2_ film, including the MIT hysteresis loop steepness, possible defect-related metastable states and others, which are defined by preparation protocol of VO_2_ and by crystalline properties of a substrate. Meanwhile, accurate control of minor temperature alterations due to the photothermal effect in hybrid Au-VO_2_ metasurfaces with complicated topology is still challenging. Note that comprehensive control of a device’s temperature and conductivity of its elements and an accurately chosen applied light intensity within addressing hologram are the key parameters to obtain advanced optical addressing of a metasurface. Once these issues are resolved, the design of a gradient metasurface with pixels switched to more than two states will become doable. As a result, the quality of several important applications, including focusing, polarization rotation, modulation and beam steering, would be improved. Indeed, the exact control of the temperature immediately on the surface, which can be different from the one at the back side of the sample or the heater surface, is a key important issue. Thus, in the case of optically triggered metasurfaces, the temperature data feedback should be obligatory in-situ, taken into account to optimally apply the needed laser light power to completely or partially switch the metasurface elements.

There are several methods to monitor the temperature of nanoparticle arrays or even individual nanoobjects [18,19,20,21,22,23]. Each method has some benefits and drawbacks in compatibility towards the object to characterize, achievable temperature resolution, range of working temperature, laboriousness, reliability, cost-efficiency, etc. These conditions may seriously limit the applicability of a particular method to study some specific type of object. For example, using thermocouples or IR thermometry, one typically may obtain averaged and not accurate results if special efforts were not undertaken. Among the more reliable optical methods, one should name luminescence nanothermometry [18], exploiting emission from quantum dots or a ruby substrate with incorporated Cr^3+^ ions [19], Raman thermometry [20], as well as more delicate techniques such as holographic interferometry [21,22] or photothermal imaging [23].

We propose a new method that exploits surface acoustic waves (SAW) to simultaneously accurately monitor the altered temperature and electric conductivity of a hybrid Au particle-VO_2_ composite film heated and exposed by laser light. The studied Au particle/VO_2_/TiO_2_ multi-layer structures are deposited on the YX128° cut LiNbO_3_ piezoelectric crystal where a SAW at 120 MHz may propagate—the surface heating results in the SAW velocity alteration, which is observed as the S_21_ phase shift. The SAW phase shift is calibrated as a function of the bare LiNbO_3_ substrate temperature and is further used to monitor crystal surface temperature variation induced by any process. In particular, we monitor laser-assisted heating induced by plasmonic particles. Additionally, the temperature-triggered alteration of sheet conductivity in VO_2_ film due to MIT is in situ controlled by monitoring a SAW attenuation change.

In this study, we quantitatively characterize the photothermal effect in the macroscopic array of Au nanostars at varied surface coverage and in the monolayer of Au coupled particles deposited on VO_2_/TiO_2_/LiNbO_3_ multi-layer structure as well as on the bare LiNbO_3_ substrate. Furthermore, an optically triggered insulator-metal transition assisted by the photothermal effect was studied at varied light power. On a broader scale, the proposed SAW-based method is expected to become a new cost-effective tool to accurately monitor surface temperature and sheet conductivity of phase change metasurfaces. It is more important for testing and optimizing their elements during design. Also, we believe that the method will be useful in studies of the various heat, electric or/and mechanical stress-related phenomena in nanosystems for nanoelectronics, photonics, metamaterials, plasmonics, terranostics and other fields.

## 2. Materials and Methods

### 2.1. Pulsed Laser Deposition of VO_2_/TiO_2_ on Piezoelectric LiNbO_3_ Substrates

VO_2_/TiO_2_ films of 10 × 10 mm in Samples C, D and E were deposited on the central part of 10 × 20 mm YX-128° cut LiNbO_3_ substrates using a pulsed laser deposition method. Radiation of KrF laser (at 248 nm, 10 Hz) was focused to a 2 × 5 mm square spot on a rotating VO_2_ or TiO_2_ targets to give a laser fluence of 2 J/cm^2^. The target to substrate distance was set to be 5 cm. First, a ~70 nm TiO_2_ buffer layer was deposited on LiNbO_3_ substrate at 640 °C in oxygen ambient at 3 × 10^−2^ mbar for 2000 laser shots. Next, a ~130 nm VO_2_ film was deposited at 560 °C an oxygen pressure of 2 × 10^−2^ mbar.

### 2.2. Raman Spectroscopy and XPS Study of VO_2_ Film

Raman spectra of VO_2_/TiO_2_ films (Sample D) were studied using Renishaw inVia Reflex Raman spectrometer with a spectral resolution better than 1 cm^−1^. The samples were excited by the light of the Ar+ laser at 514 nm with a power density less than using ×50 long focal distance objective (NA = 0.5). Raman scattering was collected in the backward direction. Crossed polarizers were not used. The incident laser power was minimized to avoid local sample heating.

X-ray photoelectron spectroscopy (XPS) experiments were carried out using Thermo Scientific ESCALAB 250 system with a monochromatic Al-Kα X-ray source. The energy resolution is 0.6 eV, as found from the observation of the Ag 3d_5/2_ line. The sample was exposed with an X-ray beam 500 microns in diameter. The binding energies were obtained using a calibration line of C 1 s at 285.0 eV.

### 2.3. Fabrication of Quiasi-Homoheneous Monolayers of Au Nanostars

First, 15 nm spherical Au seeds were prepared by adding 3.1 mL 1% sodium citrate to 100 mL of boiling 0.01% chloroauric acid on a magnetic stirrer (700 rpm). Within 15 min, the color of the solution changed from colorless to orange–red. The final Au concentration was 1.6 × 10^12^ mL^−1^. Next, 1 mL of polyvinylpirrolidone (10 kDa, 100 mg/mL in ethanol) was added to 100 mL of Au nanoparticles under stirring (700 rpm). The mixture was allowed to react for 1 h. The PVP-coated Au seeds were centrifuged twice (12,000 rpm, 45 min), the supernatant was discarded, and the particles were dispersed in 4 mL of ethanol to achieve a final Au concentration of 4 × 10^13^ mL^−1^. Next, AuNSt were prepared following a modification of the standard PVP/DMF approach [24]. First, 10 g of PVP was dissolved in 100 mL of DMF. Next, an aqueous solution of 0.2 mL 120 mM HAuCl_4_ was added, followed by 0.2 mL of seeds. Within 150 min, the color of the solutions changed from colorless to deep blue, indicating the formation of AuNSt. Finally, the AuNSt were cleaned via centrifugation (7500 rpm, 20 min) and resuspended in 20 mL of H_2_O. The final concentration of nanostars was 4 × 10^11^ mL^−1^, corresponding to optical density 8 at the plasmon resonance wavelength (as measured in the cuvette of 1 cm length).

A representative TEM image of as-prepared nanostars is shown in Figure 1b. The particles have an average size of about 70 nm and consist of a 30–40 nm core accompanied by 5–10 tips. This particle structure results in the main plasmon peak at a wavelength of 820 nm, as shown in Figure 1a.

The LiNbO_3_ substrates were cleaned with ethanol and immersed into 0.1% polivinylpiridine (PVPir, 160 kDa, 0.1% ethanol solution) for 3 h. Then, samples were washed with ethanol and water. VO_2_/TiO_2_/LiNbO_3_ substrates were then placed into nanostars solutions with particle concentrations of 4, 1 and 0.5 × 10^11^ mL^−1^ to obtain Sample C–E correspondingly. Bare LiNbO_3_ substrate and the nanostars solutions with particle concentration of 2 × 10^11^ mL^−1^ were used to get Sample A. The samples were incubated for 4 h to promote nanoparticle adsorption. After incubation, substrates were washed with water, dried and stored at ambient conditions until use.

### 2.4. PLD of a Monolayer of Au Plasmonic Coupled Nanoparticles

A quasi-homogeneous self-assembled macroscopic monolayer of 5–10 nm Au coupled plasmonic nanoparticles (Sample B) shown in Figure 1d was prepared using the PLD method in Ar ambient at 0.7 m bar for 400 laser shots.

Due to some distribution of size, ellipticity and inter-particle separating gaps in the array of coupled nanoparticles, the observed absorption band is significantly broadened as it is a superposition of many plasmonic bands of individual particles. For ~10 nm particles, the scattering absorption is predominant, and the input of light scattering is minor. As a result, the transmission spectrum reveals a characteristic broad band near ~780 nm and a shoulder in the blue/UV range of ~250–450 nm, as shown in Figure 1c.

### 2.5. Characterization and Measurements

We use the setup shown in Figure 2a to in-situ monitor the samples’ surface temperature and conductivity change. Surface acoustic wave (SAW) is excited in the samples deposited on the LiNbO_3_ substrate. Two YX-128° cut LiNbO_3_ substrates with a one-directional inter-digital transducer (IDT) on each piece were used to generate and detect a SAW. We use the standard LiNbO_3_ YX128° cut, a reasonably widely available LiNbO_3_ crystal cut on the market, which gives the optimal effective electromechanical coupling coefficient and, thus, allows one to obtain the lowest SAW propagation losses without additional efforts. Furthermore, IDTs are designed to have a central frequency of 120 MHz and an aperture of 1.2 mm. Thus, the generated SAW propagates in the acoustic channel of the same width. Therefore the channel width where SAW is propagating is also only 1.2 mm or a bit smaller.

However, the sample width is 1 cm. So, we probe the properties only in the central part of the sample, which is exposed with a laser beam 1 mm diameter. By changing the beam spot size, we experimentally found that beam size corresponds to the largest signal (SAW phase alteration). Note that with too tight focusing, the heat flow rate to unexposed areas increases as the temperature gradient increases. Thus, we have chosen the beam spot close to the SAW channel width. The examined sample is mounted and tightly pressed on the top of these two substrates near their edges in a “surface to surface” position so that a SAW generated by the first IDT may readily propagate via all three LiNbO_3_ substrates and be detected by the second IDT. A SAW S_21_ phase shift and attenuation are measured with a network analyzer Obzor 103 as a function of sample temperature while direct heating or/and of a laser power when heating with light. A photothermal effect in nanoparticles induced by laser radiation and MIT in VO_2_ film is accurately characterized in this manner. Previously, a simplified version of this setup was used by us to monitor the conductivity alteration in ZnO and Zn(Mg,Al)O films under deep UV/UV radiation exposure [25,26].

## 3. Results

### 3.1. Setup Calibration Procedure

First, a temperature-induced phase shift of the SAW at 120 MHz, propagating in bare YX-128° cut LiNbO_3_ substrate, was characterized. The temperature of the substrate was controlled in the range of 30–80 °C with an accuracy of ~0.1 °C by using a home-built Peltier heater. The SAW S_21_ phase shifts were measured at varied temperatures, as shown in Figure 2b. SAW attenuation is not altered when the LiNbO_3_ crystal is heated, as its conductivity remains constant. This linear SAW phase shift versus temperature dependence shown in Figure 2c with the SAW phase shift of 20.7 induced by temperature change for every 1 °C is used in further experiments as a calibration plot to accurately monitor the temperature of the samples’ surface in the further studies.

### 3.2. SAW to Probe Photothermal Effect in Monolayers of Single Nanostars and Monolayer of Au Coupled Nanoparticles

One may monitor the average temperature of any nanoobjects or film deposited onto the LiNbO_3_ crystal by using the calibration curve discussed above. Note that in the more general case, the temperature alteration may be induced by any physical or chemical phenomena. In particular, this allows us to characterize a photothermal effect induced in hybrid plasmonic systems when exposed to laser radiation.

Laser light at 808 nm was focused to a 1 mm round spot on the sample’s surface in the central part where the SAW is propagating, as shown in Figure 2a. The light power density was controlled in the range of 2.7–27 W/cm^2^ by using pulse width modulated electric triggering of the CW laser. The SAW phase shift and the corresponding temperature increase (obtained from the calibration curve) on the surface of Sample A with Au nanostars and Sample B with a monolayer of Au-coupled nanoparticle on bare LiNbO_3_ substrate is shown in Figure 2d. The temperature increases by 3.7 °C and 5 °C in the case of samples A and B, correspondingly when exposed to 25 W/cm^2^ laser light. Note that the temperature of a bare LiNbO_3_ substrate is increased to only ~1.5 °C when similar light exposure is used, as shown with open rectangles in Figure 3d. The laser-induced heating of the bare substrate is occurred due to nonzero optical absorption in LiNbO_3_ crystal. Optical transmission of LiNbO_3_ crystal in the visible/near IR range is shown in Figure 1c.

### 3.3. SAW to Probe the Temperature Induced MIT in “Au Nanostars/VO_2_” System

The temperature-induced metal-to-insulator transition in samples with AuNSts/VO_2_/TiO_2_ structures deposited on the LiNbO_3_ substrate with varied Au nanostars coverage (Samples C–E) was characterized by monitoring both the SAW attenuation and the phase shift as shown Figure 3.

When the insulator-to-metal transition in VO_2_ film occurs at 74 °C, the SAW attenuation is decreased from 70 dB to 51 dB at 88 °C due to sheet resistance alteration, as shown in Figure 3a,b. Note that SAW attenuation in VO_2_/TiO_2_/LiNbO_3_ sample is increased from 62 dB to 70 dB upon the deposition of Au nanostars. SAW attenuation in Sample E as a function of temperature for heating and cooling regimes is shown in Figure 3b. Both the SAW phase shifts and SAW attenuation were measured at varied temperatures simultaneously in Samples C–E. Upon complete thermal balancing, the sample surface temperature obtained from the SAW phase shift data revealed a great correlation with the value measured by the thermo-resistive sensor, which was mounted on the heater near the back side of the sample. However, when measuring both values immediately after increasing the heating power, i.e., before thermalization, the temperature on the surface revealed a somewhat lower value compared to one of the sensors and became equal upon some delay.

Next, Samples C–E were exposed to 808 nm laser light at varied power densities instead of direct heating. The obtained SAW phase shifts (right *y* axis), as well as the corresponding temperature (left *y* axis), are shown in Figure 3c. The photothermal effect, i.e., the surface temperature increase due to light-induced light heating of Au nanostars, was characterized in this manner. Samples C–E temperature is increased up to 3.5 or 5 °C per different amounts of nanostars deposited on the surface, as shown in Figure 3c. Indeed, it is quite apparent that the samples with a larger amount of nanostars show a greater photothermal effect.

### 3.4. SAW to Probe the MIT Induced by Combination of Heating and Light Exposure

Finally, the combined switching of AuNSts/VO_2_/TiO_2_ structures was studied. Sample E was heated to the temperature of 74 °C just at the beginning of MIT. Next, the sample was exposed to laser light at 808 nm with power density controlled between 12.5 and 23.4 W/cm^2^. The laser power was controlled by the PWM triggering of a cw laser. Laser “ON/OFF” cycles were chosen to be 10 min, enough for guaranteed stabilization of VO_2_ temperature and conductivity. As the VO_2_ film grown on TiO_2_/LiNbO_3_ substrate has not abrupt but quite gradual MIT dynamics, one should increase the temperature to ~11 °C to completely switch the film to a metallic state (see Figure 3b). However, when exposed to light irradiance lower than ~23 W/cm^2^, the photothermal effect induced by plasmonic particles results in a temperature increase of ~3 °C or less. Therefore a complete switching of this VO_2_ film at such applied laser power does not occur. Instead, the film may be switched to one of the metastable states defined by incident light irradiance, as shown in Figure 3d. Namely, once Sample E is exposed to 12.5 W/cm^2^ light, the temperature is slightly increased in accordance with the corresponding curve in Figure 3c, and the SAW attenuation decreases from 68 dB to 67 dB claiming increased film conductivity. Upon switching off the additional heating light, VO_2_ is returned to the initial conductivity state. If the sample is heated with larger light power, for instance, with 14.7 W/cm^2^, the SAW attenuation is decreased from 68 dB to 66.5 dB. Still, it does not return to 68 dB even after 10 min, claiming that film conductivity is increased and remains in some intermediate metastable state. A similar scenario with metastable states is observed for all cases when exposing VO_2_ to light at higher power, as shown in Figure 3d.

The conductivity changes in VO_2_ result in the altered SAW attenuation range of 68 dB and 63.5 dB. Therefore, the state of partially switched VO_2_ film can be accurately controlled.

### 3.5. Composition and Lattice Vibrations of VO_2_ Films on TiO_2_/LiNbO_3_ Substartes

To better understand the origin of unusual metastable states in VO_2_ films prepared on TiO_2_/XY-128° LiNbO_3_ substrates, the samples were studied with X-ray photoelectron spectroscopy and Raman spectroscopy at altered temperature.

A typical XPS spectrum of VO_2_/TiO_2_ film used in Samples C–E is shown in Figure 1e. XPS spectra reveal V 2p_3/2_, V 2p_1/2_ and O 1s peaks. V 2p_3/2_, V 2p_1/2_ mountains reveal a composition of V^4+^ and V^5+^ oxidation states. Namely, the more substantial peaks at 516.7 eV and 524.3 eV were assigned to be V 2p_3/2_, V 2p_1/2_ of V^5+^ oxidation state, and weaker peaks at 515.4 eV and 522.4 eV do correspond to V 2p_3/2_ and V 2p_1/2_ of V^4+^ oxidation state [27,28]. The VO_2_ phase in the V^5+^ oxidation state is ~3.7 times greater than the VO_2_ phase in the V^4+^ state. The O 1 s peak does also reveal two components. The less intense peak at 529.6 eV corresponds to oxygen bound to vanadium atoms in V^4+^ and V^5+^ oxidation states [27,28]. The stronger peak at 531.6 eV likely corresponds to oxygen [29] or/and OH groups [30,31] adsorbed on the surface. Some authors also assign this peak as a state corresponding to oxygen vacancies inside the film [31]. However, further efforts should be made to come to such a conclusion. The film is completely oxidized, and no traces of metallic vanadium are detected (not shown here).

Raman spectra of VO_2_/TiO_2_ film on LiNbO_3_ substrate at 30 °C shown in Figure 1f reveal Ag^n^ vibration modes at 191 and 224 cm^−1^ of a monocline (M1) VO_2_ lattice and a mode near 264 cm^−1^ which is assigned to monocline (M2) phase of VO_2_ [10,32]. However, a mode at 264 cm^−1^ is often erroneously assigned to the M1 phase [32,33]. However, phase M1 is disappeared from spectra already at 65–70 °C, while traces of M2 mode are still detected even at a higher temperature of 90 °C. In addition, two extra narrow modes near 405–409 cm^−1^ and 420 cm^−1^ (not identified yet) appear at a temperature of 85–90 °C when MIT occurred. Due to several very intense and broad vibration modes of monocrystalline LiNbO_3_ substrate, other VO_2_ and TiO_2_ lattice modes could not be detected.

## 4. Discussion

### 4.1. On SAW to Probe Photothermal Effect in Nanoparticles

The nonzero imaginary part of the dielectric function of gold results in great energy dissipation upon the light absorbed at wavelengths of nanoparticles plasmon in the neat IR range. Photo-induced heating of a sample surface occurs. This corresponds to the first scenario of temperature increase. Besides, the electric field is tightly localized in the near field of a particle, typically no larger than ~10 nm from the particle surface. The electric field localization known as a “hot spot” is strongly dependent on the particle’s geometry and plasmonic coupling, if any. Suppose the medium which surrounds the particle is also capable of absorbing incident radiation. In that case, a “hot spot” may also induce extra heating, the second possible reason for the temperature increase.

In the studied samples with isolated or weekly coupled particles, the impact of the second plasmonic fields-related scenario seems much smaller compared to the first one, in which the light absorption in nanoparticles results in energy dissipation and further heat transfer to VO_2_. Indeed, the comparison of the laser-induced surface temperature increase in Sample A in Figure 2d with AuNSts and in Samples C–E with AuNSts/VO_2_ in Figure 3c does not reveal a significant difference. As the AuNSts in Samples A and C–E are contacting with LiNbO_3_ and VO_2_, correspondingly, and the light absorption efficiency of these materials is different, the laser-induced heating of these surfaces should also be different if the plasmonic field effect would be strong. However, it is not the case here, evidencing that the plasmonic field effect is minor.

Despite the evidence that the plasmonic field-induced absorption mechanism in VO_2_ is minor in discussed samples, it does not mean it could not be significant in other plasmonic systems. We expect that in strongly coupled plasmonic systems, the impact of the “hot spots” related temperature increase channel could be much more pronounced. Also, it is notable that in VO_2_ with abrupt MIT, the thermal or plasmonic non-thermal effects, if any, do show a much more dramatic effect compared to those systems where VO_2_ has a slowly evolving transition and minor temperature increase induced by these effects is not able to trigger a complete transition. Besides, in typical VO_2_/c-Al_2_O_3_ films, the MIT evolves dramatically to the final metallic state once it has been triggered, which often makes it challenging to follow the process, especially if the temperature of the sample surface is not accurately monitored in situ.

Let’s briefly compare the main features of the proposed SAW-based method with several popular methods to study the temperature of individual nanoobjects or the entire surface. Luminescence nanothermometry typically reveals small spectral shifts of a broad luminescence spectrum of quantum dots for only 0.065 nm per °C, which defines quite a low accuracy which is not enough for characterizing small temperature alterations [34]. However, when monitoring a temperature-induced luminescence quenching rate change which for some QDs may approach 1% per °C, one may obtain a moderate temperature resolution between 0.2 and 1 °C [18]. The more accurate measurements with 0.1 °C achieved uncertainty are still quite rear for this approach [35]. Also, special corrections related to light-induced self-heating of nanothermometers should be taken into account to avoid errors. Besides, the measurement range should not exceed the temperature of QDs degradation, which is often not too high. Raman thermometry based on measuring of intensity, width or shifts of Stokes and anti-Stokes Raman modes does typically show shifts for 0.027 cm^−1^ per °C [20], which are also too small to resolve the temperature alterations of 0.1 °C as a typical Raman spectrometer has a spectral resolution of 1 cm^−1^ or worse. However, the method allows one to monitor very high temperatures, typically up to 700 °C, which makes it effective when studying nanosystems for thermoplasmonics [20]. The photothermal imaging method is based on monitoring local alterations of the refraction index near nanoobjects heated by laser light [23]. Providing sub-diffraction resolution, the method best suits studying photothermal characteristics of single absorbing species at the nanoscale but not temperature distribution at large surfaces. Holographic interferometry is best suited for studying thick and transparent samples with large thermal expansion coefficients where the refractive index is significantly altered with temperature [22,36]. Note that thermo-optic coefficients of materials may be different for several orders of magnitude. For instance, the coefficients dn/dT for the fused silica and PMMA are roughly equal to 8.46 × 10^−6^ °C^−1^ and 1.1 × 10^−4^ °C^−1^, correspondingly [36]. If the temperature-sensitive substrate material is properly chosen, one may obtain a high-temperature resolution of ±0.1 °C [21] or even as high as ±0.0085 °C [36].

The photothermal effect in Samples A and B with plasmonic particles is several times greater than in a bare LiNbO_3_ substrate which allows us to clearly distinguish the photo-induced heating effect exclusively associated with the plasmonic system (see Figure 2d). It should be emphasized that the LiNbO_3_ temperature alteration for only 5 °C corresponds to a great SAW phase shift for ~100 degrees. Taking into account electronic noise, we estimate the temperature measurement precision of our SAW-based technique to be not worse than 0.2 °C with an uncertainty of ~0.1 °C when using wave at 120 MHz. It should be emphasized that if using SAW at a higher frequency, one would achieve even higher temperature resolution. Summarizing the above discussion, it should be claimed that the proposed SAW-based method is comparable or even better in temperature resolution and reliability compared with the best existing techniques, including optical interferometry [22], fluorescence nanothermometry [18] and Raman thermometry [20]. However, it is much more cost-effective and much less labor-consuming. More accurate direct comparison of the possibilities of proposed SAW based on optical interferometry and fluorescence nanothermometry as the most suitable alternatives is a subject for further study.

### 4.2. On SAW to Probe Metal to Insulator Transition VO_2_

The SAW velocity change (Δυ) and wave intensity attenuation (Γ) are induced by acoustoelectric interaction with a film which is given by [37]:(1)$$\frac{\Delta \upsilon }{\upsilon_{0}}=\frac{k_{ef}^{2}}{2}\frac{1}{1+\sigma_{sh}{}^{2}/\sigma_{M}^{2}}$$
(2)$$\Gamma =k_{ef}^{2}\left(\frac{2\pi }{\lambda }\right)\frac{\sigma_{sh}/\sigma_{M}^{}}{1+\sigma_{sh}{}^{2}/\sigma_{M}^{2}}$$
where $\upsilon_{0}$, *λ*, *σ_sh_*, $k_{ef}^{2}\approx 0.056$ and *σ_M_* = $\upsilon_{0}$(*ε_s_ + ε*_0_) = 1.48 × 10^−6^ (Ω/◻)^−1^ denote a SAW initial velocity in LiNbO_3_ (3880 m/s), wavelength, sheet conductivity of VO_2_ layer, effective electromechanical coupling coefficient and the material constant called Maxwell conductivity, respectively. Dielectric permittivity of LiNbO_3_ substrate *ε_s_* = 3.7 × 10^−10^ F·m^−1^. When the sheet conductivity of the VO_2_ film is limited to Maxwell conductivity *σ_M_,* the attenuation Γ of a SAW approaches a maximum.

Indeed, the initial sheet conductivity *σ_sh_* of the prepared VO_2_ film at room temperature may be smaller or greater than Maxwell conductivity *σ_M_*. When the initial film conductivity *σ_sh_* is greater than *σ_M_* and is increased due to the MIT in VO_2_ film, the attenuation Γ is decreased. However, if the initial conductivity of a VO_2_ film prepared in some special conditions is smaller than *σ_M_,* the attenuation could grow with increased conductivity. More information about similar phenomena may be found in our previous reports on SAW propagating in Zn(Mg, Al)O and ZnO films [25,26].

Also, the SAW velocity and, thus, the measured phase of arriving SAW are significantly dependent on surface temperature, and the dependence of a SAW phase on VO_2_ sheet conductivity is minor.

### 4.3. The Differences of MIT in VO_2_/Al_2_O_3_ and VO_2_/TiO_2_/LiNbO_3_ Films

Remarkably, the VO_2_/TiO_2_ films grown on LiNbO_3_ with gradual switching and intermediate metastable states along the MIT curve at which one may stabilize film conductivity are very different from traditional VO_2_ films grown on the Al_2_O_3_ substrates which we studied before [10]. Indeed, traditional VO_2_/Al2O_3_ films typically show abrupt MIT once the process is triggered. Moreover, one can’t stabilize the conductivity at any state within the isolator to the metal transition curve. Also, we do typically observe MIT in VO_2_/Al_2_O_3_ films when the film is pre-heated by current to the beginning of the transition and additionally exposed with light at an irradiance of 0.4 W/cm^2^ to 0.28 W/cm^2^ (at 4 mm laser spot) [10] which is at least ~100 times lower compared to those needed for complete switching of VO_2_/TiO_2_/LiNbO_3_. The difference between irradiances can be explained by gradual MIT dynamics (~11 °C per transition) and larger heat flow at smaller laser spots.

In our previous Raman study of VO_2_/Al_2_O_3_ films [10], we already reported on a mixture of M1 and M2 monocline lattices and their similar temperature behavior when the M1 phase disappeared after MIT at ~67 °C, and the M2 phase was not affected by temperature change [10].

Note that the VO_2_/TiO_2_/LiNbO_3_ films in the present study show MIT at 74 °C, which is higher than in VO_2_/Al_2_O_3_ films at 67 °C [10]. Indeed, the observed shifts of Ag_n_ vibrations indicate the presence of mechanical stresses in the VO_2_ lattice. The correlation of MIT shift and mechanical stresses in VO_2_/TiO_2_ films was reported by several authors. However, a related MIT is typically shifted to a lower temperature [38,39]. Therefore, YX-128° cut LiNbO_3_ substrate likely results in mechanically induced MIT shift to a higher temperature.

XPS study reveals a mixture of phases with a ratio of V_2_O_5_:VO_2_ = 3.7 and likely existing oxygen vacancies in the VO_2_ film, which is very different from the typical VO_2_ films prepared [32,40] where the VO_2_ phase is prevailing. Remarkable that a similar phase ratio transformation towards a greater V_2_O_5_ amount was reported for films initially prepared at 500 °C by the e-beam evaporation method, which was further thermally oxidized at temperatures of 400–550 °C [33]. Note that the V_2_O_5_ phase has an MIT at 280° [41] and can hardly be related to MIT at 74 °C or the memory effect observed in samples. However, a large V_2_O_5_ phase and XPS peak at 531.6 eV, possibly related to oxygen vacancies in bulk or on the film surface, claims that some other oxygen-related reactions during the film design have occurred. We believe these formed oxygen-related features are responsible for the unusual memory effect in VO_2_/TiO_2_ films. A detailed study of this phenomenon is to be continued. We believe that the proposed new type of VO_2_ film grown on TiO_2_/LiNbO_3_ with gradual MIT and memory effect in metastable conductivity states may be used in metasurfaces where gradient characteristics are desired, but slow tuning is not an issue.

## 5. Conclusions

In conclusion, we propose a new method based on surface acoustic waves to accurately monitor a photothermal effect in plasmonic nanosystems and plasmon-enhanced MIT in VO_2_-based composites. Measuring a 120 MHz SAW phase shift in temperature-unstable LiNbO_3_ substrate, we acquire the surface temperature data with a resolution of ±0.1 °C. Simultaneously, by probing the SAW attenuation, one may monitor the conductivity change during the MIT in VO_2_. A monolayer of isolated Au nanostars and coupled Au nanoparticles boost the photothermal effect, which increases the VO_2_ surface temperature by 5 °C and 3.7 °C, respectively, when exposing the sample to laser light at 25 W/cm^2^.

The VO_2_/TiO_2_/LiNbO_3_ films reveal an option to stabilize their conductivity at any point within the MIT curve and accurately tune it by increasing incident light power which is very different from the behavior of similar films prepared on Al_2_O_3_ or TiO_2_ substrates. We believe that this unique memory effect of the prepared structures will be demanding for designing tunable middle IR/THz optical devices, namely mirrors with controlled reflection, frequency filters, metalens with tunable focus, gradient gratings and others. Furthermore, in a broader scale, the proposed SAW-based method may be placed in one raw with the best available optical methods for temperature monitoring, including optical interferometry, fluorescence nanothermometry and Raman thermometry. It will be useful for characterizing the photothermal effect in hybrid nanosystems, metasurfaces and optoelectronic devices.

## Figures and Tables

**Figure 1 materials-16-02579-f001:**
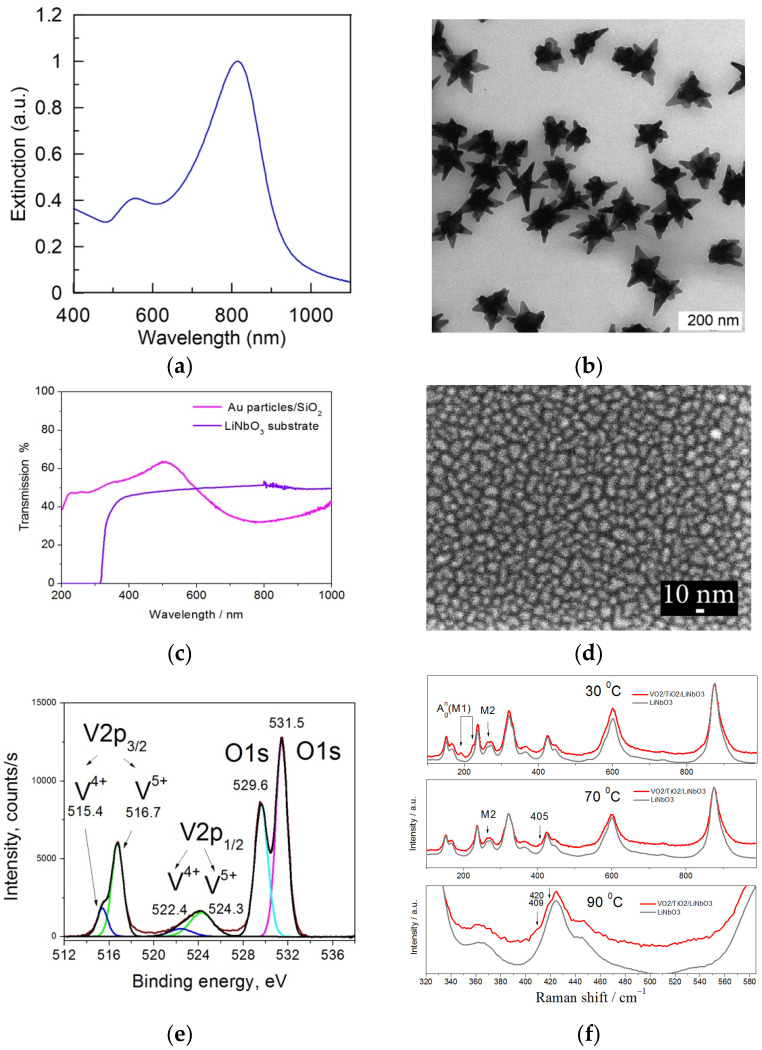
(**a**) Normalized extinction spectra of AuNSts used in Samples A, C–E; (**b**) representative TEM image of the nanostars; (**c**) optical transmission spectrum of Au coupled plasmonic particles similar to those deposited in sample Sample B and transmission of bare YX-128°-cut LiNbO_3_ double-side polished substrate; (**d**) SEM image of Au coupled plasmonic particles in Sample B; (**e**) XPS spectra of VO_2_ films on the TiO_2_/LiNbO_3_ substrate. The fitting of the spectrum with multiple peak functions is shown by colored curves; (**f**) Raman spectra of Sample D at different temperature taken in the area of VO_2_/TiO_2_ film and of a bare LiNbO_3_ substrate.

**Figure 2 materials-16-02579-f002:**
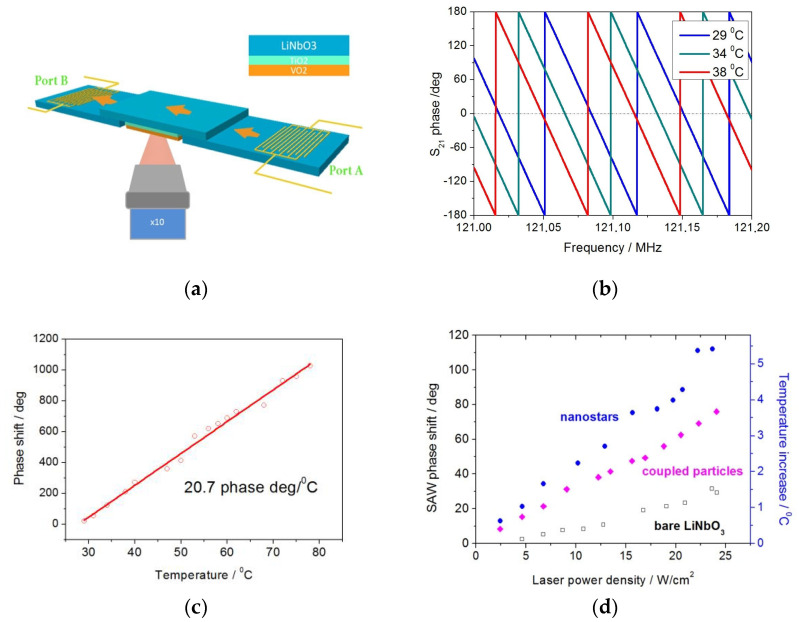
Sketch of the experimental setup (**a**); S_21_ phase of SAW in bare YX-128° cut LiNbO_3_ substrate shifted with increased temperature (**b**) and corresponding linear dependence of a 120 MHz SAW phase shift as a function of temperature (**c**); SAW phase shift (left *y*-axis) and corresponding substrate temperature increase (right *y* axis) induced by 808 nm cw laser at a varied power density in samples with a monolayer of Au nanostars (Sample A) and coupled Au particles (Sample B) deposited on LiNbO_3_ substrate. Corresponding values measured on bare LiNbO_3_ substrate are shown with open squares (**d**).

**Figure 3 materials-16-02579-f003:**
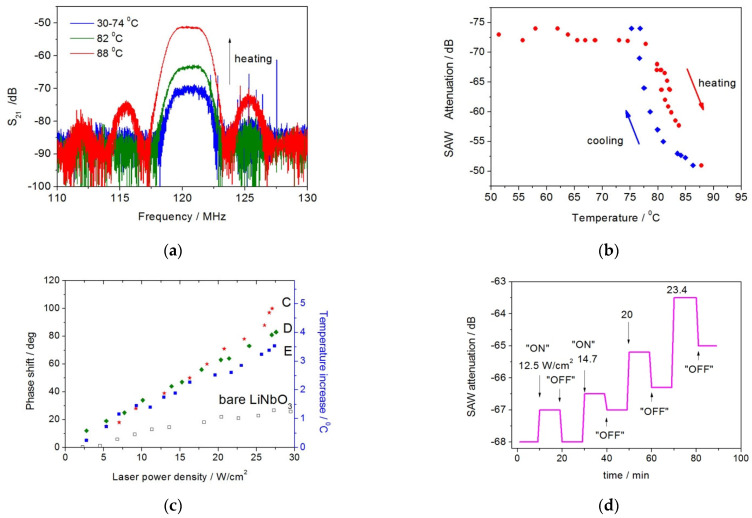
S_21_ characteristics of AuNSts/VO_2_/TiO_2_ structure (Sample E) during MIT triggered by temperature (**a**); altered SAW attenuation in Sample E while temperature triggered MIT. Sample heating and cooling are shown with red and blue points respectively (**b**); SAW phase shift and corresponding surface temperature of Samples C–E and bare LiNbO_3_ substrate when exposed with incident 808 nm laser light at different power density (**c**); SAW attenuation in Sample E measured at applied constant thermal heating up to 74 °C and additional 808 nm light exposure at varied (increased) power. “ON/OFF” corresponds to light switching on/off (**d**).

## Data Availability

Not applicable.

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
