# Peer review of "Photothermal Effect and Phase Transition in VO2 Enhanced by Plasmonic Particles"

_materials, 2023, doi:10.3390/ma16072579_

Round 1
Reviewer 1 Report
In paper, “Photothermal effect and phase transition in VO2 enhanced by
plasmonic particles” the authors proposed and demonstrated a new approach to simultaneously probe both the altered temperature and electric conductivity of a hybrid Au particle-VO2 film composite by monitoring a phase shift and an attenuation of surface acoustic wave in YX128° cut LiNbO3 substrate. It is a complete work, well written, well organized, including interesting results. All sections such as introduction, experimental arrangement, and results were adequately carried out. I think the manuscript reports an acceptable amount of information for a scientific paper. I recommend to manuscript to be accepted in Materials Journal in its present form.
Author Response
Thank you for appreciating our manuscriptReviewer 2 Report
The manuscript presents an interesting study on attempting to use of plasmonic particles to enhance the photothermal effect and phase transition of VO2. The photo-induced temperature change is measured to a very high degree of resolution with their new method based on SAW phase shift. Overall, the manuscript is a valuable contribution to the field of photothermal effects and phase transitions in VO2 enhanced by plasmonic particles. With some improvements addressing my following comments, this manuscript has the potential to be published on this journal.
1. One of the selling points of this study is the plasmon-enhanced photothermal effect. VO2 film itself has too high transmission rate and low photo-to-therm conversion rate. My question is how you distinguish the photo-thermal effect from the pure “thermal” effect. After all, the Au particles can absorb the light energy and then conduct the thermal heat to the VO2 film. If this is the case, dark film like carbon deposition amorphous carbon can do even better job. Do you have any evidence that the surface plasmon enhanced radiation field directly interact with VO2 film via some non-thermal effects?
2. The beam size is much smaller than the film size which renders temperature non-uniformity. How does this effect influence your measurement of the SAW phase shift?
3. In line 311, it should be "not worse" instead of "not worth". Please improve the axis label and add legend to Figure 3(c)
Author Response
- One of the selling points of this study is the plasmon-enhanced photothermal effect. VO2 film itself has too high transmission rate and low photo-to-therm conversion rate. My question is how you distinguish the photo-thermal effect from the pure “thermal” effect. After all, the Au particles can absorb the light energy and then conduct the thermal heat to the VO2 film. If this is the case, dark film like carbon deposition amorphous carbon can do even better job. Do you have any evidence that the surface plasmon enhanced radiation field directly interact with VO2 film via some non-thermal effects?
Reply:
We are thankful to reviewer for this very important comment. Indeed, the photo-thermal and the pure thermal effects are difficult to be distinguished as they both finally result in a temperature change. You are absolutely right that VO2 itself has very low light to heat conversion efficiency. After the deposition of nanoparticles there are two scenarios possible: (1) photothermal heating of nanoparticles, i.e. absorption of light by particles with further energy dissipation to heat. The heat is then transferred to VO2 due to thermal conductivity; (2) plasmon induced electric near field enhancement in the ~10 nm neighborhood to the particle may induce extra light absorption in VO2.
Our current findings suggest that the second scenario where plasmon enhanced radiation field directly interacts with VO2 is much less probable, in particular in the studied samples, compared to the first scenario based on plasmonic absorption by nanoparticles and further thermal heating of VO2. Indeed, by comparing the laser induced heating effects in the sample A with AuNSts only and in the samples C-E with AuNSts/VO2 one would not reveal significant difference (see Fig. 2(d) and Fig. 3(c) for more details). Namely, the similarity of surface temperature in samples with nanostars deposited immediately onto the LiNbO3 and on top of VO2/TiO2/LiNbO3 structure results in nearly identical measured responses (SAW phase shift) when exposing the samples with laser light at altered power density. As the AuNSts are contacting with LiNbO3 and VO2 in these cases, which have different light absorption efficiency the heating of these surfaces should be also different if the plasmonic field effect would be strong. However, it is not a case here evidencing that plasmonic field effect is minor.
However, we believe that it is too early now to make some claims about absence of a plasmon induced extra absorption in VO2. We hope that in other systems this effect could be much more noticeable. In particular, in more complicated strongly coupled plasmonic systems one may expect a larger impact of high electric fields near the “hot spots” which will enhance light absorption in surface of VO2, however the volume with elevated electric fields is quite small compared to film volume. Probably, this will best work in systems with large surfaces covered with “hot spots” or where the “hot spots” are spread in the film bulk.
Also, it is notable, that in VO2 with abrupt MIT the thermal or plasmonic non-thermal effects, if any, do show much more dramatic effect compared to those systems where VO2 has a slowly evolving transition and minor temperature increase induced by these effects is not able to trigger a complete transition. Besides, in a typical VO2/с-Al2O3 films the MIT is evolving dramatically to the final metallic state once it has been triggered, which makes it challenging to follow the process especially if the temperature of the sample surface is not accurately monitored in situ.
Concerning remark on a deposition of any structure with perfect or very high absorption to boost optical absorption, further heat transfer to VO2 and finally a heating/switching of VO2, you are completely right. We sure, that will be one of the most efficient ways to boost such materials. Indeed, many different systems including carbon based structures, plasmonic systems (anodized aluminum film covered with Au for instance) or others will work better compared to our isolated nanostars or monolayer of weekly coupled nanoparticles used in this paper. However, we should emphasize here that we used the simplest available plasmonic nanosystems to illustrate the fact that our SAW based method is capable to probe such effects and to characterize the possibilities of the method. The systems with optimal light absorption efficiency will be the topic of our further study.
Changes made to the manuscript:
In the studied samples with isolated or weekly coupled particles an impact of the second plasmonic fields related scenario seems to be much smaller compared to the first one in which the light absorption in nanoparticles results in energy dissipation and further heat transfer to VO2. Indeed, the comparison of the laser induced surface temperature increase in Sample A in Figure 2 (d) with AuNSts and in Samples C-E with AuNSts/VO2 in Figure 3 (c) does not reveal a significant difference. As the AuNSts in Samples A and C-E are contacting with LiNbO3 and VO2, correspondingly, and the light absorption efficiency of these materials is different the laser induced heating of these surfaces should be also different if the plasmonic field effect would be strong. However, it is not a case here evidencing that plasmonic field effect is minor.
In spite of the evidence that the plasmonic field induced absorption mechanism in VO2 is minor in discussed samples, it does not mean that it could not be significant in other plasmonic systems. We do expect that in strongly coupled plasmonic systems the impact of the “hot spots” related temperature increase channel could be much more pronounced. Also, it is notable, that in VO2 with abrupt MIT the thermal or plasmonic non-thermal effects, if any, do show much more dramatic effect compared to those systems where VO2 has a slowly evolving transition and minor temperature increase induced by these effects is not able to trigger a complete transition. Besides, in a typical VO2/с-Al2O3 films the MIT is evolving dramatically to the final metallic state once it has been triggered which often makes it challenging to follow the process especially if the temperature of the sample surface is not accurately monitored in situ.
- The beam size is much smaller than the film size which renders temperature non-uniformity. How does this effect influence your measurement of the SAW phase shift?
Reply:
We are very thankful to reviewer for this very important remark. Indeed, the apertures of the IDTs which generate and receive a SAW are 1.2 mm width. We have forgotten to mention that fact in manuscript. Now it is added.
Changes made to the manuscript:
IDTs are designed to have a central frequency of 120 MHz and aperture of 1.2 mm, thus, the generated SAW is propagating in acoustic channel of the same width. Therefore the channel width where SAW is propagating is also only 1.2 mm or a bit smaller. However, the sample width is 1 cm. So, we probe the properties only in the central part of the sample, which is exposed with laser beam 1 mm diameter. By changing the beam spot size we experimentally found that beam size which corresponds to largest signal (SAW phase alteration). Note, that with too tight focusing the heat flow rate to unexposed areas is increased as the temperature gradient is increased, thus we have chosen the beam spot to be very close to the SAW channel width.
- In line 311, it should be "not worse" instead of "not worth". Please improve the axis label and add legend to Figure 3(c)
Reply: Thanks for pointing to these minors. These minors were corrected.
Changes made to the manuscript:
…we estimate the temperature resolution of our SAW based technique to be not worse than ±0.1 ° C when using wave at 120 MHz
In caption to Figure 3 (c):
…SAW phase shift and corresponding surface temperature of Samples C,D and E and bare LiNbO3 substrate when exposed with incident 808nm laser light at different power density(c)
Reviewer 3 Report
The article "Photothermal effect and phase transition in VO2 enhanced by 2 plasmonic particles" is of outstanding quality in all aspects.
The structure of the paper corresponds to the structure expected from scientific articles. The English of the paper is good, with no spelling mistakes or typos. The text of the article is well written and understandable, there are no logical jumps, it is clearly written.
The introduction provides a good overview of the research field as a whole and also of the immediate background to the problem under consideration and related research. There are no gaps in the description of the literature.
The second section describes very precisely and clearly the materials and procedures used. The description even gives the reader the possibility to repeat the study. Such precise descriptions are rare in scientific articles.
Section 3 presents the results. The description is adequate, the figures are easy to follow, contain the necessary information, the fonts used are appropriate and the captions are clear. The results presented are sufficiently detailed to support the message of the article.
Section 4 analyses the results and compares them with other methods and results obtained by other authors.
The conclusion of the article is well supported by the results presented.
I recommend publication of the article unchanged.
Author Response
Thank you for appreciating our workReviewer 4 Report
In this manuscript, authors have proposed an interesting method based on surface acoustic waves to accurately monitor a photothermal effect in plasmonic nanosystems. I would recommend the publication after a minor revision.
- "There are several methods to monitor temperature of nanoparticle arrays or even in- 72 dividual nanoobjects." Add references
- would authors confirm YX128°?
- if photothermal effect could be measured using optical nanothermometry, and establish a relation?
-
Author Response
- There are several methods to monitor temperature of nanoparticle arrays or even individual nanoobjects." Add references
Changes made to the manuscript:
There are several methods to monitor temperature of nanoparticle arrays or even individual nanoobjects [19-24].
- would authors confirm YX128°?
Reply:
We use the standard LiNbO3 YX128° cut, which is a fairly widely available LiNbO3 crystal cut on the market. Indeed, this special cut allows one to obtain a material with a maximal effective electromechanical coupling coefficient of k^2=0.056 and, thus, minimal SAW losses (lowest attenuation). When the cut is a bit different, the effective electromechanical coupling coefficient and SAW attenuation are altered dramatically, which is readily experimentally detected. We also evaluated the electro-mechanic constant of our LiNbO3 substrate. It is indeed, 0.056 similar to the data in handbook for this crystal cut. Also, we experimentally measured the SAW velocity [see our previous papers Smart Mater. Struct. 2017, 26, 035029; Smart Mater. and Struct. 2019, 28, 065024] and obtained the values which are all very consistent with theoretical estimation for this substrate.
Changes made to the manuscript:
We use the standard LiNbO3 YX128° cut, which is a fairly widely available LiNbO3 crystal cut on the market, which gives the optimal effective electromechanical coupling coefficient and, thus, allows one to obtain the lowest SAW propagation losses without additional efforts.
- if photothermal effect could be measured using optical nanothermometry, and establish a relation?
Reply:
Luminescence nanothermometry typically reveals small spectral shifts of a broad luminescence spectrum of quantum dots for only 0.065 nm per °C which defines quite low accuracy which is not enough for characterizing small temperature alterations [28]. However, when monitoring a temperature induced luminescence quenching rate change which for some QDs may approach to 1% per °C one may obtain a moderate temperature resolution between 0.2 and 1 °C [19]. The more accurate measurements with 0.1°C achieved uncertainty of are still quite rear for this approach [29]. Also, special corrections related to light induced self-heating of nanothermometers should be taken into account to avoid errors.
Indeed, the proposed SAW based method seems to have better accuracy of temperature measurements, especially when the SAW is generated at higher frequency (2.4 GHz). We are planning to prepare such ships ant test them in near future. We also are planning to compare the possibilities of both methods (luminescence nanothermometry and SAW based) in further study. This step will need extra efforts with building an additional setup.
Changes made to the manuscript:
More accurate direct comparison of the possibilities of proposed SAW based with optical interferometry and fluorescence nanothermometry as most suitable alternatives is a subject for further study.
Reviewer 5 Report
The authors investigated the photothermal effect busing SAM, where samples are the Au nanostars, the Au particles on the surfaces, and VO2 films. The results about Au nanostars and Au particles revealed the plasmonic coupling effect. VO2 results indicated that MIT effect was observed. Interestingly, there is a possibility of the existence of the metastable states in VO2 films. Furthermore, they claimed that the this method is superior to the other conventional methods. These results are very nice and interesting not only in terms of the measurement method but also in terms of the new findings. I think that I can easily expect plasmonic effect. However, as for the VO2 films, the authors found that there is no recover of SAM attenuation, which is interpreted as the existence of metastable states. This is unpredictable and interesting scientifically. Then, this topic is suitable to this journal. However, there are some concerns about the temperature and discussion, as shown below. In conclusion, this manuscript could meet the criteria for the publication if the manuscript is properly revised by answering the comments. My comments are written below.
(1) The authors obtained the relationships between SAW phase shift and temperature (for example, Figure 2). This temperature is for the surface of LiNbO3 substrate SAW is going through, isn’t it? So I wonder if this temperature is really the same as the those of Au nanostars, Au particles and VO2 films. This point should be commented in this manuscript although the authors might wrote this point in the previous studies.
(2) What is the precision and uncertainty of measured temperature? Although the resolution is mentioned when the substrate has high sensitivity, these things should also be discussed.
(3)The authors claimed that film conductivity remaining is interpreted as the existence of intermediate metastable. This is one of the most interesting points in this manuscript. So, the following thins are added.
(3-1)VO2 films quality should be clarified. Polycrystals, domain size, composition and so on.
(3-2) MIT about VO2 film has long history and there are many researches. Recently, it was reported that metastable states (M1,T, M2, R) in MIT was observed by TEM in 2023, depending on the strain. >From these preceding results, possible metastable states should be discussed.
(4) Minors In p.6, the authors mentioned Figure 3(d) and talk about SAW phase shifts and temperature. Maybe, this figure number is wrong isn’t it?
Author Response
- The authors obtained the relationships between SAW phase shift and temperature (for example, Figure 2). This temperature is for the surface of LiNbO3 substrate SAW is going through, isn’t it? So I wonder if this temperature is really the same as the those of Au nanostars, Au particles and VO2 films. This point should be commented in this manuscript although the authors might wrote this point in the previous studies.
Reply: We are thankful to reviewer for this remark. The SAW at 120 MHz is penetrating to the sample to the distance of ~33 microns from the surface. Thus, the SAW velocity (detected phase shift) depends heavily on the temperature of this LiNbO3 layer. As the nanoparticles are located on the surface of ~200 nm VO2/TiO2 film which has quite high thermal conductivity the temperature of the stationary heated particles (with continuous laser) should be quite close to the temperature of the LiNbO3 surface. The similarity of surface temperature in samples with nanostars deposited immediately onto the LiNbO3 and on top of VO2/TiO2/LiNbO3 structure results in nearly identical measured responses (SAW phase shift) when exposing the samples with laser light at altered power density.
Changes made to the manuscript:
As the AuNSts in Samples A and C-E are contacting with LiNbO3 and VO2, correspondingly, and the light absorption efficiency of these materials is different the laser induced heating of these surfaces should be also different if the plasmonic field effect would be strong. However, it is not a case here evidencing that plasmonic field effect is minor.
- What is the precision and uncertainty of measured temperature? Although the resolution is mentioned when the substrate has high sensitivity, these things should also be discussed.
Reply: We are thankful to reviewer for this important remark. We did additional estimations of electronic noise and came to a bit more accurate conclusion.
Changes made to the manuscript:
Taking into account electronic noise we estimate the temperature measurement precision of our SAW based technique to be not worse than 0.2 ° C with uncertainty of ~0.1° C when using wave at 120 MHz.
(3)The authors claimed that film conductivity remaining is interpreted as the existence of intermediate metastable. This is one of the most interesting points in this manuscript. So, the following thins are added.
(3-1)VO2 films quality should be clarified. Polycrystals, domain size, composition and so on.
(3-2) MIT about VO2 film has long history and there are many researches. Recently, it was reported that metastable states (M1,T, M2, R) in MIT was observed by TEM in 2023, depending on the strain. >From these preceding results, possible metastable states should be discussed.
Reply: We are thankful to reviewer for this important comment. Indeed, metastable sates found in studied VO2/TiO2/LiNbO3 samples are of significant interest and need to be studied in more detail in future special study concerning solid state physics of this type of VO2. We should emphasize here that the present work is dedicated more to the new SAW based method and its novel possibilities for characterization of metasurfaces, plasmonic systems, ect.
However, we tried to do our best (in frame of our current possibilities) to shed more light to the new structural and composition properties of studied samples. In particular, we studied temperature dependence of Raman and XPS spectra.
Changes made to the manuscript:
2.2 Raman spectroscopy and XPS study of VO2 film
Raman spectra of VO2/TiO2 films (Sample D) were studied using Renishaw inVia Reflex Raman spectrometer with spectral resolution better than 1 cm-1. The samples were excited by light of Ar+ laser at 514 nm with power density less than using ×50 long focal distance objective (NA=0.5). Raman scattering was collected in the backward direction, crossed polarizers were not used. The incident laser power was minimized to avoid local sample heating.
X-ray photoelectron spectroscopy (XPS) experiments were carried out using ESCALAB 250 system with monochromatic Al-Kα X-ray source. Energy resolution is 0.6 eV as found from observation of Ag 3d5/2 line. The sample was exposed with x-ray beam 500 microns diameter. The binding energies were obtained using a calibration line of C 1s at 285.0 eV.
3.5 Composition and lattice vibrations of VO2 films on TiO2/LiNbO3 substartes
To better understand the origin of unusual metastable states in VO2 films prepared on TiO2/XY-128° LiNbO3 substrates the samples were studied with X-ray photoelectron spectroscopy and Raman spectroscopy at altered temperature.
A typical XPS spectrum of VO2/TiO2 film used in Samples C-E is shown in Figure 1(e). XPS spectra reveal V 2p3/2, V 2p1/2 and O 1s peaks. Both V 2p3/2, V 2p1/2 peaks reveal a composition of V4+ and V5+ oxidation states. Namely, the stronger peaks at 516.7 eV and 524.3 eV were assigned to be V 2p3/2 and V 2p1/2 of V5+ oxidation state and weaker peaks at 515.4 eV and 522.4 eV do correspond to V 2p3/2 and V 2p1/2 of V4+ oxidation state [32,33]. The amount of VO2 phase in V5+ oxidation state is ~3.7 times greater than VO2 phase in V4+ state. The O 1s peak does also reveal two components. The less intense peak at 529.6 eV corresponds to oxygen bound to vanadium atoms in V4+ and V5+ oxidation states [32,33]. The stronger peak at 531.6 eV is likely corresponding to oxygen [34] or/and OH groups [35,36] adsorbed on the surface. Some authors do also assign this peak as a state, corresponding to oxygen vacancies inside the film [36]. However, further efforts should be done to come to such a conclusion. The film is completely oxidized and no traces of metallic vanadium are detected (not shown here).
Raman spectra of VO2/TiO2 film on LiNbO3 substrate at 30 °C shown in Figure 1(f) reveal Agn vibration modes at 191 and 224 cm-1 of a monocline (M1) VO2 lattice and a mode near 264 cm-1 which is assigned to monocline (M2) phase of VO2 [11 ,37]. However, often a mode at 264 cm-1 is erroneously assigned to M1 phase [37, 39]. However, the phase M1 is disappeared from spectra already at 65-70 °C, while traces of M2 mode are still detected even at higher temperature of 90 °C. In addition, two extra narrow modes near 405-409 cm-1 and 420 cm-1 (not identified yet) are appeared at temperature of 85-90 °C when MIT is occured. Due to several very intense and broad vibration modes of monocrystalline LiNbO3 substrate other modes of VO2 and TiO2 lattice could not be detected.
In discussion:
In our previous Raman study of VO2/Al2O3 films [11] we already reported on mixture of M1 and M2 monocline lattices and their similar temperature behavior when M1 phase is disappeared after MIT at ~67 °C and M2 phase was not effected by temperature change [11].
Note, that the VO2/TiO2/LiNbO3 films in present study do show MIT at 74 °C which is higher compared to the one in VO2/Al2O3 films at 67 °C [11]. Indeed, the observed shifts of Agn vibrations indicate the presence of mechanical stresses in the VO2 lattice. The correlation of MIT shift and mechanical stresses in VO2/TiO2 films was reported by several authors, however a related MIT is typically shifted to lower temperature [40,41]. YX-128° cut LiNbO3 substrate likely results in mechanically induced MIT shift to higher temperature.
XPS study reveals a mixture of phases with ratio of V2O5:VO2=3.7 and likely existing oxygen vacancies in the VO2 film which is very different from the typical VO2 films prepared [37, 38] where VO2 phase is prevailing. Remarkable, that similar phase ratio transformation towards greater V2O5 amount was reported for films initially prepared at 500 °C by e-beam evaporation method which were further thermally oxidized at temperatures of 400-550 °C [39]. Note, that V2O5 phase has a MIT at 280° [42] and hardly can be related to MIT at 74 °C or memory effect observed in samples. However, large amount of V2O5 phase together with XPS peak at 531.6 eV, which is possibly related to oxygen vacancies in the bulk or on film surface claim that some additional oxygen related reactions during the film design has occurred. We believe that these formed oxygen related features are responsible for unusual memory effect in VO2/TiO2 films. A detailed study of this phenomenon to be continue.
4) Minors In p.6, the authors mentioned Figure 3(d) and talk about SAW phase shifts and temperature. Maybe, this figure number is wrong isn’t it?
Reply: Many thanks for pointing to this minor. Instead of Figure 3 (d) one should read Figure 2 (d).
Changes made to the manuscript:
The SAW phase shift and the corresponding temperature increase (obtained from calibration curve) on the surface of the Sample A with Au nanostars and the Sample B with monolayer of Au coupled nanoparticle on bare LiNbO3 substrate is shown in Figure 2 (d).
Round 2
Reviewer 2 Report
I am satisfied with the revised version and recommend to publish it in the present form.
Reviewer 5 Report
The manuscript is properly revised. This is suitable to the publication.